# Vitamin D Status and Its Determinants in a Paediatric Population in Norway

**DOI:** 10.3390/nu12051385

**Published:** 2020-05-12

**Authors:** Mads N Holten-Andersen, Johanne Haugen, Ingvild Oma, Tor A Strand

**Affiliations:** 1Department of Paediatrics, Lillehammer Hospital, Innlandet Hospital Trust, 2626 Lillehammer, Norway; 2Department of Clinical Medicine, Faculty of Medicine, University of Oslo, 0316 Oslo, Norway; 3Department of Medical Microbiology, Innlandet Hospital Trust, 2626 Lillehammer, Norway; Johanne.Haugen@sykehuset-innlandet.no (J.H.); ingvild.oma@gmail.com (I.O.); 4Centre for Intervention Science in Maternal and Child Health, Centre for International Health, University of Bergen, P.O. Box 7800, 7803 Bergen, Norway; tors@me.com; 5Department of Research, Innlandet Hospital Trust, 2381 Brumunddal, Norway

**Keywords:** vitamin D, 25(OH)D, child, adolescent, age, ethnicity

## Abstract

Recommendations for sufficient vitamin D intake in children were recently revised in Norway. However, optimal levels of vitamin D are still debated and knowledge on supplementation and vitamin D levels in healthy children in Norway is scarce. Therefore, we measured the plasma-concentration of 25-hydroxyvitamin D (25(OH)D) in children and adolescents attending the outpatient paediatric clinics in Innlandet Hospital Trust, Norway during two consecutive years (2015–2017). We recruited 301 children and adolescents aged 5 months to 18 years (mean 7.8, SD 4.4 years) for the study and obtained sample material for 25(OH)D measurements from 295 (98%). Information on diet, vitamin D supplementation, sun exposure, ethnicity, parental education and general health was collected by questionnaire. 25(OH)D levels were analysed and determinants for 25(OH)D were estimated by linear regression. 1.0% of the children had deficient levels (25(OH)D < 25 nmol/L) and 21.0% had insufficient levels (25–50 nmol/L). 25(OH)D levels ranging from 50 to 75 nmol/L were found among 38.3%, while 39.7% had levels above 75 nmol/L. The mean 25(OH)D level was 70.0 nmol/L (SD 23.4, range 17–142 nmol/L) with a significant seasonal variation with lowest levels in mid-winter and highest in late summer. In addition to seasonal variation independent determinants for 25(OH)D-levels were age of the child, parental ethnicity, vitamin D supplementation and soda consumption. Along with parental ethnicity other than Nordic, age was the strongest determinant of 25(OH)D, with adolescents having the lowest levels.

## 1. Introduction

Although a well-defined threshold value for 25-hydroxyvitamin D (25(OH)D) is lacking, it is established that severe vitamin D deficiency increases the risk for hypocalcaemia and impaired bone health in children, such as rickets and osteomalacia [1]. In recent years there has been an increased interest in measuring vitamin D for other indications as well, most likely as a result of indications of possible associations between mild vitamin D deficiency and other health outcomes, such as immunologic, infectious, cardiovascular, and cancer diseases in epidemiological studies [2]. Despite decades of research on vitamin D, questions still remain unanswered regarding mechanisms and possible causality in the putative non-calcemic effects of vitamin D [2], resulting in an on-going debate on optimal vitamin D status and modalities for vitamin D measurement. In the latest Nordic guidelines (from 2012) for intake of nutrients, the recommended intake of vitamin D was increased for children above 2 years of age in order to target a sufficient vitamin D concentration, defined as 25(OH)D > 50 nmol/L [3]. In 2008, a systematic review of existing studies of 25(OH)D status among healthy Norwegians in different age groups was published [4]. For children and adolescents existing studies were relatively few and most were characterized by older date, small sample size or limited age span (infants, toddlers, adolescents) [4]. Earlier studies from the 1980s have reported a high prevalence of vitamin D deficiency among exclusively breastfed infants in Norway [5] as well as vitamin D insufficiency among older children and adolescents at the end of the winter season [6]. A more recent study from Tromsø (69° N) in Northern Norway including 890 adolescents aged 15–18 years reported that 25(OH)D deficiency or insufficiency was prevalent among 60.2% of the participant and more frequent in boys [7]. Determinant for 25(OH)D among the adolescents included vitamin supplementation, physical activity, sunbathing on holidays and use of solarium [7]. Still, detailed knowledge on current 25(OH)D levels in children and adolescents in different age groups in Norway is lacking. Furthermore, detailed information on factors with potential influence on 25(OH)D levels in Norwegian children and adolescents including seasonal variation, diet, vitamin D supplementation, sun-exposure and ethnicity is needed. In this paper, we describe 25(OH)D levels and possible determinants in a paediatric population, consisting of children and adolescents aged 5 months–18 years, attending the outpatient paediatric clinics at Innlandet Hospital Trust, Norway, during two consecutive years (2015–2017).

## 2. Materials and Methods

### 2.1. Study Design and Enrolment of Study Subjects

This cross-sectional study was undertaken at the paediatric departments of Innlandet Health Trust from January 2015 until March 2017. The study had two objectives: () to evaluate the association between 25(OH)D levels and susceptibility of lower respiratory tract infection in children less than five years of age using case-control design, and (2) to survey 25(OH)D levels in children and adolescents in the Innlandet region, Norway. The sample size for the cross sectional component assumed a proportion of deficiency of 50%. For an absolute precision of 5% (95% CI) we needed a sample of nearly 400. Data from the case-control study have not yet been published and the case component, consisting of children hospitalized because of acute lower respiratory tract infection, will not be further discussed in this paper. The participants in the control group were enrolled at the paediatric outpatient clinics at Lillehammer, Gjøvik, and Elverum (61° N), Norway. Eligible for inclusion were 5 months–18 years old children and adolescents attending the outpatient clinics. Prerequisites for inclusion were informed written parental consent and the child had to have a blood test taken as part of the paediatric evaluation. Fulfilment of inclusion criteria was at the discretion of the attending paediatrician at the paediatric outpatient clinic. Exclusion criteria included presence of acute lower respiratory tract infection or other acute infection, chronic severe liver, kidney, heart or lung disease, chronic severe metabolic or nutritional disorders or oncologic disease. 40.9% (123 of 301) of the participants in the control group had a non-excluding chronic disease. A variety of diagnoses were represented including congenital heart disease, allergies, neurologic, nephrologic, gastrointestinal, endocrine and dermatologic disorders, with the most frequent being type 1 diabetes (21) and asthma (24). An individual project number was assigned to each patient, which along with date of inclusion, date of birth, and gender were registered. Ethical clearances and approval of consent procedure were obtained from the Regional Committee for Medical and Health Research Ethics in the southeast part of Norway, REC Southeast (2015/199/REK sør-øst B). Informed written consent was obtained from at least one parent of all recruited children. The implementation of all aspects of the projects was in agreement with the International Ethical Guidelines for Research Involving Human Subjects as stated in the latest version of the Helsinki Declaration.

### 2.2. Background Data and Anthropometrics

Age and gender were registered for all children and adolescents enrolled in the study. Anthropometrics, including height/length (cm) and weight (kg), were measured by a paediatric nurse in the outpatient clinic on the day of inclusion using standardized methods and scales. Height was measured to the nearest millimeter and weight to the nearest 100 g [8]. Supplemental health information was obtained using a questionnaire completed by parents of enrolled children. Children above 12 years of age were encouraged to complete the questionnaire with their parents.

### 2.3. Questionnaire

Questions included parental education (9 years of primary school, 9 years of primary school + 1–2 years high school, 9 years of primary school + 3 years high school, higher education college, higher education university), current parental employment (fulltime job (>30 h/week), part time job (<30 h/week), unemployed/seeking job, student, working at home, disability support, other), numbers of smokers in the household (none, 1, 2, 3, >3), parental ethnicity (maternal and paternal country of origin), asthma of the child (not having asthma, mild asthma, moderate asthma, uses asthma medication when needed, uses asthma medication daily, does not use asthma medication), vitamin D supplementation (daily, 4–6 times/week, 1–3 times/week, 1–3 times/month, seldom/never), intake of fatty fish, vegetables, fruit, sweets, and soda drinks (daily, 4–6 times/week, 1–3 times/week, 1–2 times/fortnight, 1–2 times/month, seldom/never), daily time spent outside with sun exposure of the skin during the summer months (a lot, moderate, little, almost never), use of sunscreen (always, often, seldom, never), and previous lower respiratory tract infection requiring antibiotic treatment (yes, no). In accordance with our “a priori” plan for interpretation of the data some of the descriptive ordinal variables were dichotomized in order to perform meaningful analyses. Vitamin D supplementation was dichotomized to yes/no: no including “seldom/never” and yes including remaining categories. Dietary intake of fatty fish was dichotomized to yes/no: no including seldom/never and yes including remaining categories. Intake of sweets and soda were dichotomized to weekly/not weekly: weekly including “daily”, “4–6 times/week” and “1–3 times/week”, not weekly including remaining categories. Education of mother/father was dichotomized to higher/lower education: “higher education” including college and university with the remaining groups designated “lower education”. Parental ethnicity was categorized into the following three groups: both parents of Nordic origin, one parent of Nordic origin, no parent of Nordic origin. Smoking in the household and asthma were dichotomized to yes/no. Sun-exposure was dichotomized as a lot/little: those answering “a lot” in one category and the remaining (moderate, little, almost never) into a group designate “little”. Sunscreen use was dichotomized to yes/no: those using sunscreen always and often in one category (“yes”) and those reporting use seldom and never in the other (“no”). Different categories of dichotomization for both vitamin D supplementation and fatty fish intake were explored but were not found to affect the results of the regression analyses.

### 2.4. Blood Samples and Laboratory Analyses

Blood (0.5 mL plasma) was drawn as capillary samples in the infants and toddlers and by venipuncture in the older children. Plasma was separated from blood cells by centrifugation at 2500 G followed by storage at 4 °C until analysis the same day. P-25(OH)D-concentration was measured using the protein binding assay Roche Elecsys vitamin D total at the Department of Laboratory Medicine at Innlandet Health Trust along with other blood-test parameters requested by the doctor involved according to hospital routines. The Roche Elecsys vitamin D total assay has a functional sensitivity below 15 nmol/L and within-run CVs of ≤6% and between-run CVs of ≤8% [9]. Month of the year at which the blood sampling had occurred was registered for all recruited children. In linear regression analyses month of the year was dichotomized into season; summer May–October, winter November–April.

### 2.5. Definition of Vitamin D Status

In this paper we use the cut-off 25(OH)D > 50 nmol/L for vitamin D sufficiency, which is in line with the 5th edition of the Nordic Nutrition Recommendations [10]. This cut-off also corresponds to the “Global Consensus Recommendations on Prevention and Management of Nutritional Rickets” [11] from 2016, the Institute of Medicine (IOM) from 2011 [12], and The Paediatric Endocrine Society from 2008 [13]. To reflect another commonly used cut-off for vitamin D insufficiency referred to in the literature, we also report the cut-off < 75 nmol/L [14]. 25(OH)D levels between 25 and 50 were thus regarded as insufficiency of vitamin D in our study. Definitions of deficiency also vary. In the Nordic countries vitamin D deficiency is defined as 25(OH)D < 25 nmol/L, which is therefore used in this publication. A cut-off of 30 nmol/L is often used in other European countries, whereas others use a cut-off at 50 nmol/L [14]. 

### 2.6. Statistical Analyses

Figures of predicted mean 25(OH)D and CI according to month of the year and age were generated using two-way fpfit-analyses. Associations between potential determinants and 25(OH)D levels were investigated in unadjusted and adjusted linear regression analyses. A step-wise manual selection process was used for evaluating determinants of 25(OH)D in multiple linear regression models [15]. The following candidate variables were included in this process: age, gender, maternal education (higher/lower), paternal education (higher/lower), parental ethnicity (both Nordic/one Nordic/none Nordic), tobacco smoke in the household (no/yes), intake of vitamin D supplements at least monthly (no/yes), intake of fatty fish at least monthly (no/yes), intake of sweets weekly (no/yes), intake of soda weekly (no/yes), sun-exposure (little/a lot), sunscreen use (no/yes), asthma (no/yes), previous infection requiring antibiotic treatment (no/yes), season of the year (summer/winter). Assessment for multicollinearity was performed using variance inflation factor (VIF). Test for interaction between parental ethnicity and vitamin D supplementation was done in additional models. Significance levels were set at 5%. Statistical analyses were carried out using SPSS Statistics for Windows, Version 23.0.0.2 Armonk, NY: IBM Corp and STATA 15.0 software (STATA).

## 3. Results

301 children were included (Table 1). Their mean age was 93.1 (SD 52.3) months corresponding to 7.8 (SD 4.4, range 0.42–17.8) years. Their mean weight was 30.2 (SD 18.1) kg and mean height 123.8 (SD 31.4) cm. For 267 (88.7%) of the children both parents had Nordic ethnicity, for 23 (7.6%) one parent had Nordic ethnicity and for 11 (3.7%) neither of the parents had Nordic ethnicity. The percentage of children receiving vitamin D supplementation was 61.9%, with 70.6% for those with one or both parents with other ethnicity than Nordic. Intake of fatty fish at least once a week occurred in 47.6%, with a higher fraction of 58.8% among those with one or both parents of non-Nordic ethnicity. Consumption of soda at least weekly was reported by 52.5%. Sun-exposure throughout the summer months was reported as “a lot” by 60.9%, and 89.2% reported use of sunscreen. Living in households where smoking was reported by 19.4% of children; 25.6% had asthma and 26.2% had previous experience of one or several episodes of lower respiratory tract infection requiring antibiotic treatment.

25(OH)D was measured in 295 children. In 6 children 25(OH)D was not measured due to too little sample material (4 children) or having been included in the study, but not having blood sample taken after all (2 children). 66% (196/295) of the participants were included during winter (November-April) corresponding to the seasonal variation in the number of children attending the out-patient clinics of Innlandet Health Trust, with the lowest numbers in mid-summer. The mean (SD) 25(OH)D-concentration was 70.0 (23.4) nmol/L ranging from 17 to 142 nmol/L (Table 2). There were substantial differences in 25(OH)D levels between the summer and the winter months with the lowest levels measured in December (mean ± SD: 57.7 ± 12.5 nmol/L) and the highest in August (mean ± SD: 96.9 ± 19.1 nmol/L). The seasonal variation of 25(OH)D levels is illustrated in Figure 1 displaying predicted mean 25(OH)D levels with 95% CI according to month of blood sampling.

Only 3 (1%) children were vitamin D deficient (25(OH)D < 25 nmol/L). All 3 children were above 12 years of age and one had parental ethnicity other than Nordic. The prevalence of vitamin D insufficiency was higher among the older children and adolescents than among the younger children (Table 3). While only 14.7% of the children below 6 years of age had insufficient levels (25(OH)D < 50 nmol/L), insufficiency (including deficiency) was present in 47.5% of the children aged 12 years or older. Both the reported intake of vitamin D supplementation and self-reported sun-exposure tended to be inversely related to age with lower prevalence among the children older than 12 years compared to the younger children with exception of sun-exposure in those <2 years of age.

Figure 2 illustrates the age-related levels of 25(OH)D with predicted mean levels and 95% CI of 25(OH)D according to age displayed. 

Age, weekly soda consumption and winter season were inversely associated with 25(OH)D, whereas vitamin D supplementation, Nordic parental ethnicity and summer season were positively associated with 25(OH)D-concentration (Table 4). Compared with children of parents where both had non-Nordic ethnicity 25(OH)D levels were 15.5 nmol/L higher in children with one parent with Nordic ethnicity and 24.5 nmol/L higher in children with both parents with Nordic ethnicity. Children who had 25(OH)D measured during summer had on average 10.2 nmol/L higher levels than those measured during winter. Children reported to use vitamin D supplementation had 8.84 nmol/L higher 25(OH)D levels on average than children not using supplementation. On average, 25(OH)D levels decreased by 1.75 nmol/L for an increase in age of 1 year. Finally, children reported to have weekly consumption of soda had on average 5.47 nmol/L lower 25(OH)D levels than those who did not. Taken together, the variables age, vitamin D supplementation, parental ethnicity, weekly soda consumption and season of the year predicted 27% of the variation in 25(OH)D-concentration in the regression analyses.

The remaining variables tested in the model (gender of the child, weekly intake of fatty fish (no/yes), parents with higher education (no/yes), self-reported sun-exposure (little/a lot), sunscreen use (no/yes), smoking in the household (no/yes), asthma (no/yes) and previous lower respiratory tract infection requiring antibiotic treatment (no/yes)) were not significantly associated with 25(OH)D, neither in the univariate nor in the multivariate analyses. No indication of multi-colliarities or significant interactions were found between tested variables.

## 4. Discussion

Only 1.0% of the children in this sample of children from Innlandet County, Norway, were vitamin D deficient (25(OH)D < 25 nmol/L), whereas 21.0% had insufficient levels of vitamin D (25(OH)D 25–50 nmol/L). Deficiency was only seen in teenagers (12–18 years). The presence of low vitamin D status was more prevalent in the teenagers than in the younger children; 42.4% of the teenagers in our study had insufficient levels of 25(OH)D, compared to 8.3%, 16.3% and 16.7% in the age-groups < 2, 2–6 and 6–12 years, respectively.

Previous data on vitamin D status in younger children in Norway is scarce. A study among 249 one-year-olds from Oslo reported a prevalence of vitamin D insufficiency of 34% during April–June 2000 for breastfed children, with a significant difference between children receiving vitamin D supplementation (14%) and those who did not (38%) [6]. Within the same study, similar levels in 227 two-year-olds were reported between March and June 2001 [6]. Almost all the children < 2 years of age received some kind of vitamin D supplementation in our study and this may be one of the reasons why vitamin D levels were higher in this population.

For the children in our study, the prevalence of deficient and insufficient vitamin D levels were very similar to multi-centre-data from the U.S. National Health and Nutrition Examination Survey (NHANES) 2001–2006. In a sample of 4558 US children (aged 1–11 years), 1% of the children were found to have vitamin D deficiency (defined as 25(OH)D < 25 nmol/L), 18% had vitamin D insufficiency (defined as 25(OH)D < 50 nmol/L), and 69% had levels < 75 nmol/L [16]. The mean 25(OH)D level was 68 nmol/L in this multi-centre study, and older age and Hispanic/non-Hispanic or black ethnicity was inversely associated with 25(OH)D-levels.

The lowest levels of 25(OH)D were found in the age-category > 12 years in our study. This finding is also consistent with previous studies. In a study from 1982, adolescents from Bergen, Norway were at risk of vitamin D insufficiency, especially in late winter. The lowest levels of vitamin D was found among 17 years old boys of which 50% were found to have insufficient levels [6]. Another population-based study from 2014 among 1038 healthy adolescents in northern Norway, reported a prevalence of vitamin D insufficiency (25(OH)D < 50 nmol/L) of 60% and a vitamin D deficiency (25(OH)D < 25 nmol/L) of 16.5% through September to April [7]. In our study, we found that age was a strong determinant of 25(OH)D, being inversely associated with vitamin D levels. This negative association is also reported in other studies: in children and adolescents (mean age ± SD, 14.5 years ± 3.1) from Oslo, Norway with excess body weight (mean BMI ± SD, 29.7 kg/m^2^ ± 6.0), Lagunova et al. reported prevalence of vitamin D insufficiency (25(OH)D < 50 nmol/L) in 24% of the 13–19 year olds but only in 4% of the 8–12 year olds [17]. Furthermore, in a recent Danish study on obesity and vitamin D status, being above the age of 14 was an independent risk factor for low vitamin D status also in the population-based control-group consisting of 2143 children and adolescents [18]. Although there is a trend towards an inverse relationship between age and sun exposure for children above 2 years of age, we did not find that self-reported sun-exposure was related to 25(OH)D levels in our study. However, we found that season of the year was a determinant for 25(OH)D levels with the lowest mean level in January and highest in August and that the prevalence of vitamin D insufficiency was highest in April and lowest in the summer months. A possible explanation for higher concentrations of vitamin D in the younger children could be a higher prevalence of vitamin D supplementation, which also was seen in our study.

An important determinant of vitamin D levels in our material was parental ethnicity other than Nordic, which was associated with on average 15.5 and 24.5 nmol/L lower concentrations of 25(OH)D when having one or both parents of non-Nordic ethnicity. The synthesis of vitamin D3 in the skin is reduced by pigmentation, and especially in higher latitudes with less sun-exposure, high pigmentation without intake of vitamin D sources, may be a risk factor for vitamin D deficiency. In the sub-group of children with parents with other background than Nordic in our study, approximately half of them (53%) were from a non-Western country and thus had a higher degree of skin-pigmentation, while the others were from different European countries. One could speculate that other habits regarding sun-exposure, clothing, food, and supplementation could be possible contributors to lower vitamin D status in this sub-group compared to the ethnic Nordic group. 

61.9% of the children in our study were given vitamin D supplementation. This was higher than a previous nationwide diet survey among Norwegian school children in 4th and 8th grade from 2015, where less than half of the 9 year olds and 13 year olds reported intake of vitamin D supplements [19]. However, supplementation was less important than both age and parental ethnicity in our regression model. Weekly soda-consumption was the only other dietary variable that was of significant importance in our study, and since a direct relationship is unlikely, this association could possibly reflect lifestyle factors with impact on sun-exposure and/or intake of vitamin D rich sources.

There are some limitations to our study. We recruited patients from the paediatric outpatient clinics at Innlandet Hospital Trust. Participants were mostly healthy children, referred to a paediatrician because of a complaint or a symptom for which no chronic disease was found. In addition, we included some children with underlying chronic illnesses like type 1 diabetes or asthma where the disease was well regulated. The aim was to recruit as healthy children as possible, but a fundamental precondition for the implementation of the project was that no blood tests should be taken without being a part of necessary or already planned investigation. Including children outside health care was therefore not optional for our study. Sun-exposure was self-reported and estimated by use of a semi-quantitative scale with five incremental levels. Sun-exposure is a challenging variable to measure precisely, and we consider the seasonal variable a better reflection of sun-exposure. In our questionnaire, we could have included a question on average daily screen-time, which possibly could have added information to the study as discussed above. Another way of addressing quantification of sun-exposure could have been to use a time-scale in the answer options with registration of hours/week; however, we speculate that such a measure would also have been imprecise. Furthermore, it could be speculated that the semi-quantitative answer alternatives in the questionnaire could be interpreted differently by the parents, and that responses could be biased by what the parents might consider to be the right answer to a given question. More detailed information on main vitamin D containing foods could have been registered using a food frequency questionnaire; however, this was not done in our study. Our results indicated limited associations between vitamin D and diet, except for vitamin D supplementation and weekly soda consumption. A general challenge for the interpretation of research results of vitamin D status is the high degree of inter-assay differences between the various methods. An underestimation of 25(OH)D was earlier reported for a similar immuno-assay [20], but to our knowledge is not known be an issue in the assay used in this study. The major strengths of this study are the relatively wide age span covering infancy to adolescence and the recruitment period (2015–2017) with inclusion and blood sampling in all months of the year allowing for evaluation of seasonal influence.

## 5. Conclusions

In this study describing 25(OH)D levels and its determinants in Norwegian children, we found that 78% of the children and adolescents enrolled had adequate levels of 25(OH)D (>50 nmol/L). 1% and 21% had deficient and insufficient 25(OH)D levels, respectively. The lowest concentrations were measured in January (mean 61.4 nmol/L) and the highest in August (mean 96.7 nmol/L). The prevalence of reported intake of vitamin D supplements was 61.9%. Age and parental ethnicity were the strongest determinants of 25(OH)D with increasing age and non-Nordic parental ethnicity predicting lower 25(OH)D levels. Vitamin D supplementation, season of the year and weekly soda consumption were also found to be independently associated with 25(OH)D levels. Prevalence of vitamin D insufficiency was highest among the adolescents and deficiency was only found in this age-group.

## Figures and Tables

**Figure 1 nutrients-12-01385-f001:**
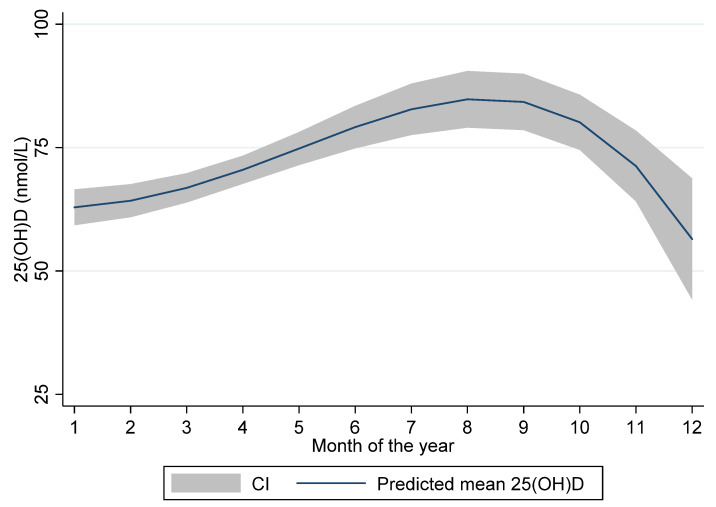
Plasma 25(OH)D levels according to month of the year in Norwegian children. Predicted mean and 95% CI during month 1 (January) to 12 (December).

**Figure 2 nutrients-12-01385-f002:**
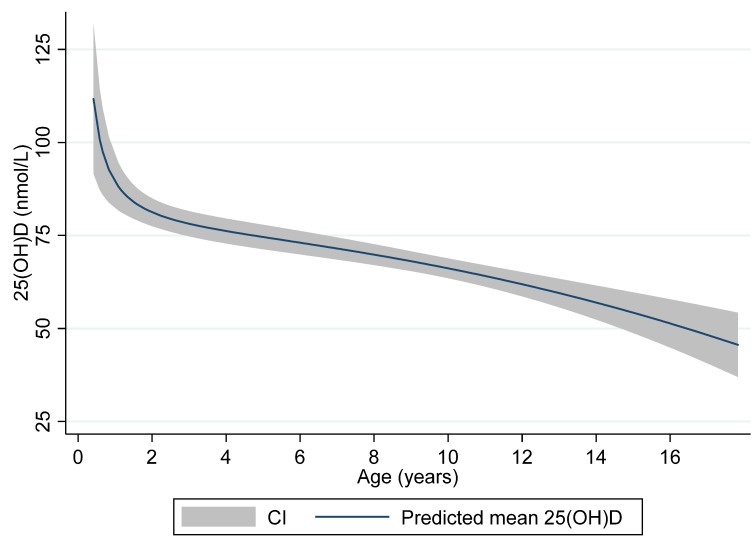
Plasma 25(OH)D levels and age. Predicted mean and 95% CI in relation to age in years.

**Table 1 nutrients-12-01385-t001:** Child characteristics.

Characteristics	Value
Age in years	7.8 (4.4)
Boys	147 (49.0)
Weight in kg	30.5 (18.2)
Height in cm	125.5 (28.8)
Supplements of vitamin D ≥ 1 monthly	182 (61.9)
Fatty fish ≥ 1 monthly	138 (47.6)
Soda consumption ≥ 1 weekly	149 (52.5)
Both parents of Nordic ethnicity	267 (88.7)
One parent with other ethnicity than Nordic	23 (7.6)
Both parents with other ethnicity than Nordic	11 (3.7)
Mother higher education	150 (52.3)
Father higher education	107 (37.9)
Sun-exposure during summer as “a lot”	173 (60.9)
Using sunscreen always/almost always/often	256 (89.2)
Smoking in the household	56 (19.4)
Asthma	73 (25.6)
ALRI requiring antibiotic treatment	75 (26.2)

Variables reported by mean (SD) or n (%). ALRI—acute lower respiratory infections. Number of *n* for each variable displayed ranged from 275 to 300 and were always above 91% of the total 301.

**Table 2 nutrients-12-01385-t002:** Plasma 25(OH)D levels by month in a population of Norwegian children.

Month	*n*	Mean 25(OH)D (SD) in nmol/L	<25 *n* (%)	25–50 *n* (%)	50–75 *n* (%)	25(OH)D ≥ 75 *n* (%)
January	60	61.4 (21.5)	2 (3.3)	16 (26.7)	29 (48.3)	13 (21.7)
February	51	67.1 (23.1)	0	15 (29.4)	20 (39.2)	16 (31.4)
March	37	68.6 (23.9)	0	9 (24.3)	15 (40.5)	13 (35.1)
April	29	64.7 (24.1)	0	11 (37.9)	7 (24.1)	11 (37.9)
May	29	72.4 (22.3)	0	5 (17.2)	9 (31.0)	15 (51.7)
June	27	82.3 (24.4)	1 (3.7)	2 (7.4)	8 (29.6)	16 (59.2)
July	5	91.6 (31.2)	0	0	1 (20)	4 (80)
August	8	96.9 (19.1)	0	0	0	8 (100)
September	9	79.4 (11.6)	0	0	3 (33.3)	6 (66.7)
October	21	73.9 (15.9)	0	1 (4.8)	10 (47.6)	10 (47.6)
November	12	76.3 (23.5)	0	1 (8.3)	6 (50.0)	5 (41.7)
December	7	57.7 (12.5)	0	2 (28.6)	5 (71.4)	0

**Table 3 nutrients-12-01385-t003:** 25(OH)D levels, vitamin D supplementation and sun-exposure by age groups.

Age	25(OH)D Mean (SD) nmol/L	25(OH)D < 25 *n* (%)	25(OH)D 25–50 *n* (%)	25(OH)D 50–75 *n* (%)	25(OH)D ≥ 75 *n* (%)	Vitamin D Supplementation *n* (%)	Sun-Exposure “a lot” *n* (%)
<2 years	89.5 (24.8) ^a^	0	2 (8.3)	3 (12.5)	19 (79.2)	22 (88.0) ^b^	7 (30.4) ^c^
2–6 years	75.8 (24.2) ^d^	0	15 (16.3)	27 (29.4)	50 (54.4)	54 (59.3) ^e^	58 (67.4) ^f^
6–12 years	68.7 (18.9) ^g^	0	20 (16.7)	64 (53.3)	36 (30)	75 (63.0) ^h^	77 (65.2) ^i^
12–18 years	55.8 (21.4) ^j^	3 (5.1)	25 (42.4)	19 (32.2)	12 (20.3)	29 (50.9) ^k^	29 (52.7) ^l^
All	70.0 (23.4) ^m^	3 (1.0)	62 (21.0)	113 (38.3)	117 (39.7)	182 (61.9) ^n^	171 (60.6) ^o^

a: *n* = 24, b: *n* = 25 c: *n* = 23, d: *n* = 92, e: *n* = 91, f: *n* = 86, g: *n* = 120, h: *n* = 119, i: *n* = 118, j: *n* = 59, k: *n* = 57, l: *n* = 55, m: *n* = 295, n: *n* = 292, o: *n* = 282.

**Table 4 nutrients-12-01385-t004:** Determinants of 25(OH)D levels in Norwegian children.

Predictor	*n*	Crude (*n* = 295)			Adjusted (*n* = 280)			
		Coeff	95% CI	*p*	Coeff	95% CI	*p*	Beta
Age (years)	295	−2.14	−2.71 to −1.57	**<0.001**	−1.75	−2.33 to −1.16	**<0.001**	−0.32
Vitamin D supplement, no		ref						
Vitamin D supplement, yes	289	10.2	4.75 to 15.7	**<0.001**	8.84	3.88 to 13.8	**0.001**	0.18
Parental ethnicity, non-Nordic		ref						
Parental ethnicity, one Nordic		10.8	−6.03 to 27.6	**0.208**	15.5	0.55 to 30.4	**0.042**	0.17
Parental ethnicity, both Nordic	295	18.8	4.77 to 32.8	**0.009**	24.5	12.1 to 36.9	**<0.001**	0.33
Soda weekly intake, no		ref						
Soda weekly intake, yes	280	−9.66	−15.1 to −4.26	**<0.001**	−5.47	−10.5 to −0.43	**0.034**	−0.12
Summer, no		ref						
Summer, yes	295	13.5	8.07 to 19.0	**<0.001**	10.2	5.14 to 15.3	**<0.001**	0.21

Age, vitamin D supplementation, parental ethnicity, soda consumption and season were significantly associated with 25(OH)D levels in both unadjusted and adjusted models. All variables in Table 4 were included in the adjusted model. Adjusted R-squared = 0.27. Coeff—unstandardized regression coefficient. Beta—standardized regression coefficient.

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
