# Peer review of "Vitamin D Status and Its Determinants in a Paediatric Population in Norway"

_nutrients, 2020, doi:10.3390/nu12051385_

Round 1
Reviewer 1 Report
This study includes 295 children aged 5 months to 18 years attending an outpatient paediatric clinic in Norway. Participants are mostly healthy, referred to a paediatrician because of a complaint/symptom for with no chronic disease was found. This is however not stated until late in the discussion and would have been very valuable much earlier, e.g. in 2.1. Also, some children with underlying chronic illnesses (DM1 and asthma, what other?) were included, but no information is given as to how many they were of the total 295.
The main outcome of interest is plasma 25(OH)D. Additional information on characteristics and possible determinants of vitamin D status was collected by a questionnaire answered by parents? or parents of children until age X and adolescents themselves from age X?.
The results on vitamin D status add to a relatively scarce literature on vitamin D status of Norwegian children and it is also relevant to report associations with possible determinants. However, I believe that the manuscript could be greatly improved, see the specific comments below:
Specific comments:
Affiliations: Affiliation 5 is not linked to an author
Abstract:
It would be valuable to give the mean (SD) age of participants in the abstract.
Lines 23-24: Describing adequate levels as 25(OH)D 50-75 nmol/L gives the reader the impression that levels above 75 nmol/L might be too high or at least not “adequate”. This sentence should be rephrased as to prevent such misunderstanding.
Introduction:
Line 38: Include reference(s).
Lines 41-44. Include reference (NNR 2012?). Also, including the year of the guidelines and recommended intake of vitamin D for children would be of benefit. Moreover, the abstract states that recommendations for sufficient vitamin D intake in children in Norway were recently revised, if those differ from the NNR it would be valuable to add more information.
Line 47: Is there any recent data at all on vitamin D status of children and adolescents in Norway? If not, then “scarce” could be changed to “none”.
Lines 46-50: This is not incorrect but may lead readers to believe that this study gives detailed knowledge on current vitamin D status in a representative Norwegian paediatric population, which is does not do.
Material and methods:
Line 55: Something published? Add reference or name of the case control study?
Line 58: Check space between 61 and N
Chapter 2.2: Why the interest in asthma and previous lower respiratory tract infection requiring antibiotics? Please include relevant literature in introduction or explain the choice of variables.
Chapter 2.2: What happens if you dichotomize vitamin D supplementation and fatty fish intake differently? 1-3 or 1-2 times/month is very little.
Chapter 2.2: You might consider interpreting the questions on sun exposure and sunscreen use together, because a child with moderate sun exposure that never uses sunscreen might be expected to get as much or more vitamin D skin synthesis than a child with lots of sun exposure that always uses sunscreen.
Line 106: Do you have a reference for the choice of summer / winter months?
Lines 110 and 124: Brackets missing for references.
Lines 115-117: You might want to add the definition of insufficient levels 25-50 nmol/L for consistency and because you use that wording in abstract.
Lines 120-121: Is this relevant information? The sample size is not close to the 400 children you mention you would need.
Line 126: In 2.2 and also in results you use 3 groups “both Nordic”, “one Nordic”, “none Nordic”. Why do you use a bivariate presentation here?
Line 127: For consistency with intakes of sweets and soda “weekly”, you might include intake of vitamin D supplements “at least monthly” and intake of fatty fish “at least monthly”.
Line 128: Might drop “self-reported” sun-exposure as all variables from questionnaire were self-reported and no need to mention once randomly.
Line 129: Might add “previous” infection
133: Please do not refer to figures that are part of the results in methods already.
Results:
Table 1 and lines 161-163: Why did you not exclude the 6 children without vitamin D measurement? The number 301 is presented for the first time in Table 1, up until then the reader believes there are only 295 children.
Table 1: I would suggest dropping the column with n for each characteristic as it gives this simple table a bit of a busy feeling. Instead you might mention in footnote that the number of n for each variable ranged from 275-300 and were always above 91% of the total 301.
Table 1: Giving age in years would be easier to understand for the reader.
Table 1: Supplements of vitamin D “≥1 per month” ?
Table 1: Here you give fatty fish at least once per week as opposed to at least once per month in methods. Please check your manuscript for consistency.
Table 1: Using sunscreen always/”often” is the term you use in methods
Table 2: One-third of children are measured during summer, the remaining during winter. Might want to mention that in discussion or elsewhere.
Table 2: For some months there are very few children. Because of the large variation in age and different ethnicity that also contribute substantially to variations in vitamin D status, you might want to consider clustering a few months together for presentation in the table. Figure 1 shows a nice difference between months.
Table 2: There is a yellow dot after February. Also, there is no need for the decimal points because of the few children in each group.
Table 3: I would suggest presenting the table before Figure 2, because when seeing the figure first the reader has no information about the age distribution, e.g. whether there are many infants and adolescents or whether they are mostly 5-10 years etc.
Lines 174 and 180 is this footnote information?
Table 3: include column for n in each age-groups. Then you could drop the busy information in footnote that gives no important information.
Lines 196-198: How many had non-Nordic parent or parents?
Table 4: The coefficient is the beta?
Table 4: What variables are the adjusted models adjusted for?
Discussion:
Lines 249 and 250: Is the latter number the SD? Fix the presentation in the parentheses.
Line 251: Rephrase the sentence “reported a prevalence (...) in 24%”
Lines 255-256: Screen-time might not be considered important for this discussion
Line 262: What is the first possible explanation?
Lines 271-275: I suggest dropping these lines as they do not add to the discussion and further distract from the main points.
Line 281: p= for the difference between ethnicity? Is this in the results or here for the first time?
Line 284: Might include this sentence in results.
Line 291: Speculations, might want to rephrase with a more cautious wording.
Lines 292-: I refer to my previous suggestion of creating a scale putting sun exposure and sunscreen use together
Line 302: There are few significant sources of vitamin D in foods, so a standardized dietary recall would be over the top, but a FFQ asking about the frequency of main vitamin containing foods would have been valuable.
Line 307-308: I would like this information much earlier.
Please check the manuscript for consistency.
Spelling: paediatric vs. pediatric.
The literature cited in introduction and discussion is very limited, among additional literature to be considered:
Vitamin D status of infants and toddlers in Oslo from 2000-2001 https://www.researchgate.net/profile/Kristin_Holvik/publication/228698999_Vitamin_D_status_in_the_Norwegian_Population/links/0046351405995907d4000000/Vitamin-D-status-in-the-Norwegian-Population.pdf
The Tromsø Study: Fit Futures, discussed and cited in Cashman et al. Vitamin D deficiency in Europe: pandemic? Am J Clin Nutr. 2016.
Itkonen et al. Vitamin D status and current policies to achieve adequate vitamin D intake in the Nordic countries. Scand J Public Health. 2020.
Reviewer 2 Report
This manuscript describes circulating vitamin D levels in Norwegian children from infants to adolescents, and factors associated with these.
The introduction could better justify why this is important for Norway in particular. I assume they get a lot less sun than many other countries, is there evidence for poor vit D status in Norway compared with other countries?
Line 51 says age range 5 months to 18 years but line 59 1 month to 18 years. Please explain. It is of interest to show the decline in vit D status over this age range but is it relevant to look at soda consumption in relation to vit D status in infants of 5 months? Although the study was probably not powered to look at age strata you could consider including age*variable interaction terms to acknowledge this.
More information on the vit D assay and quality assurance related to this is required.
It is not very useful to report average weight and height for this age range. Would be better to use growth charts and provide information on how participants compared with standards at each age.
Parents completed all the questionnaires, but adolescents aged around 18 years might know more about how much soda they drank in a week than their parents.
Dichotomising variables is not required for the analysis and losses a lot of information. This should be better justified. Similarly, for the outcome of vitamin D plasma concentration, this should be adjusted for month of sample, not just season. As the graph shows, there is a wide variation in levels over the period classified as summer.
In the conclusion minimum and maximum vitamin D concentrations do not match the figures in Table 2, where do the values come from?
Reviewer 3 Report
Manuscript ID: nutrients-783698
"Vitamin D status and its determinants in a paediatric population in Norway"
- General comments
The purpose of this study was twofold: 1) to describe the vitamin D status in paediatric population in Norway; and 2) to examine the potential factors which may predict this vitamin D status. This cross-sectional study is based on a representative sample of Norwegian children and adolescents and shed light on the factors that may predict the vitamin D status in this specific population. The manuscript is very well written and I only have minor suggestions. Regarding the vitamin D terminology, authors should be consistent. In other words, use the term ‘vitamin D levels’ or ‘25(OH)D levels’, but not both.
Regarding the methodology, authors should consider to include body mass index as a potential factor that may influence the vitamin D status. I suggest you to use obesity category as reference and cluster the rest of categories (i.e. overweight, normal weight) in non-obese predictor. Finally, authors should report the results from participants with complete data in all the assessments. It will not make the study to reduce sample size too much and the findings will be more coherent.
- Specific comments
2.1. Abstract
Line 15: Please, define 25(OH)D previously.
Line 15: Please, use ‘However’ instead of ‘But’.
Line 20: Please, follow the indications in the General comments section. Remember that we analyse vitamin D levels and report vitamin D status (i.e. deficiency, insufficiency and sufficiency).
Line 22: Please, follow the indications in the General comments section. I would write ‘21,0% showed an insufficient status (25-50 nmol/L).
2.2. Introduction
Line 38: Please, include references of epidemiological studies.
Line 49: Please, write ‘paediatric’ as in other parts of the manuscript.
2.3. Materials and Methods
Line 57: Please, indicate how many participants were enrolled, mean age and % of boys or girls.
Background data, anthropometrics and questionnaire
Authors should consider to divide this section in two paragraphs according to anthropometric assessments and questionnaire assessment.
Line 70: Please, use height instead of length. Which instruments were used to measure height and weight? Please, report the accuracy of the instruments.
Line 71: Please, calculate the BMI and classify participants into categories according to sex- and age-specific BMI cut offs.
Blood samples
Please, indicate the procedure since you obtained the samples until you analyse them and collect the data.
Please, include the kit that you used for the analyses and, the values of sensitivity and intra-assay coefficient of variation.
The definition of vitamin D status may fit better inside of the blood samples section as you only analyse this prohormone.
Sample size and statistical analyses
Please, include the sample size calculation in the ‘study design’ section.
Please, report the information of the statistical analyses in the same order that they appear in the results section. It makes easier the reading.
Line 124: in ‘multiple linear regression models 10’, what does ‘10’ mean?
Line 125: Please, include BMI as a predictor.
Line 130: Where are the interaction analyses reported?
Ethical approval
Please, include this section below the ‘study design’ section.
2.4. Results
Line 165: Please, indicate mean and SD values as you do in the Table 1: mean(SD). In fact, you can report how it is reported on the footnote of the Table 1.
Line 194: Please, report the B coefficients and the p-values for each association.
Line 196: Please, compare the 25(OH)D levels in the remaining significant variables as you do with parental ethnicity.
Line 197: Please, write ‘nmol/L’.
2.5. Discussion
Line 220: Please, include information about the other factors that were significantly associated with vitamin D levels.
Line 237: What does ‘<75 nmol/L’ mean in this study? Please, be consistent using terminology.
Line 283: Please, specify ‘(data not shown)’.
2.6. Conclusion
Please, place value on the objectives of study in order to give a clear message in your conclusion.
Line 320: Please, use vitamin D status terminology instead of vitamin D levels.
Line 321-322: Please, conclude with the ALL factors predicting vitamin D levels. Not only age and parental ethnicity.
Round 2
Reviewer 1 Report
Good work, I find the manuscript is greatly improved and valuable.
Just three minor comments this time:
Previous comment on Chapter 2.2: What happens if you dichotomize vitamin D supplementation and fatty fish intake differently? I believe that your answer to the question is valuable information to some readers, i.e. that this way of handling the data was decided "a priori" and doing it differently did not change the findings. Therefore I suggest that you add this to the manuscript.
In the footnote for Table 4: Please correct a misspelling "Tasble 4"
The sentence "In this study, we recruited patients from the paediatric outpatient clinic at Innlandet Hospital Trust." should be moved prior to the sentences starting with "Participants were mostly healthy children..." at the beginning of the limitation section.
Reviewer 2 Report
The authors have adequately addressed reviewer's comments.
Author Response
We believed that reviewer 2 does not see any need for further revisions. Please, let us know if we have misunderstood...
Sincerely, on behalf of the authors
Mads N Holten-Andersen